# A Convolutional Neural Network-Based Quantization Method for Block Compressed Sensing of Images

**DOI:** 10.3390/e26060468

**Published:** 2024-05-29

**Authors:** Jiulu Gong, Qunlin Chen, Wei Zhu, Zepeng Wang

**Affiliations:** 1School of Mechatronical Engineering, Beijing Institute of Technology, Beijing 100081, China; gongjiulu@126.com; 2North Automatic Control Technology Institute, Taiyuan 030006, China; qunlin_chen@163.com; 3Beijing Institute of Astronautical Systems Engineering, Beijing 100076, China; zhuwei9527@163.com

**Keywords:** compressed sensing, quantization, convolutional neural network, image compression

## Abstract

Block compressed sensing (BCS) is a promising method for resource-constrained image/video coding applications. However, the quantization of BCS measurements has posed a challenge, leading to significant quantization errors and encoding redundancy. In this paper, we propose a quantization method for BCS measurements using convolutional neural networks (CNN). The quantization process maps measurements to quantized data that follow a uniform distribution based on the measurements’ distribution, which aims to maximize the amount of information carried by the quantized data. The dequantization process restores the quantized data to data that conform to the measurements’ distribution. The restored data are then modified by the correlation information of the measurements drawn from the quantized data, with the goal of minimizing the quantization errors. The proposed method uses CNNs to construct quantization and dequantization processes, and the networks are trained jointly. The distribution parameters of each block are used as side information, which is quantized with 1 bit by the same method. Extensive experiments on four public datasets showed that, compared with uniform quantization and entropy coding, the proposed method can improve the PSNR by an average of 0.48 dB without using entropy coding when the compression bit rate is 0.1 bpp.

## 1. Introduction

Compressed sensing (CS) [1,2,3,4] is very suitable for resource-constrained applications due to its low complexity [5,6,7,8]. In CS, it is necessary to quantize real-valued CS measurements. The CS framework with the quantization process is called quantized compressed sensing (QCS) [8].

The QCS primarily focuses on optimizing the encoder or decoder to enhance the quality of signal reconstruction for the commonly used quantization methods. The exploration of advanced compression techniques in the realm of block-based compressed sensing (BCS) has been a focal point in contemporary research. One such advancement is the application of Differential Pulse Code Modulation (DPCM) for quantized block-based compressed sensing of images. This strategy leverages DPCM’s efficiency in exploiting spatial correlations within image blocks to minimize quantization errors, demonstrating improved bit-rate efficiency without sacrificing reconstruction quality [9]. However, the quantized measurements must undergo entropy coding to attain ideal performance. Ref. [10] proposes a progressive quantization method, which essentially improves the encoding and decoding strategies while still utilizing uniform scalar quantization. The reconstruction algorithm [11,12,13,14] has been a primary focal point in CS for optimizing the decoder.

In order to improve the quantization performance, vector quantization has been used to quantify the CS measurements [15,16]. Subsequently, to further amplify the efficiency of the vector quantization technique, ref. [17] has leveraged deep neural networks. Due to the high computational complexity of vector quantization, scalar quantization is more suitable for CS measurements. In data compression theory, the scalar quantizer that performs entropy coding on quantized output data is usually called an entropy-constrained quantizer or entropy-coded quantizer [18]. When quantization error is used as a distortion measurement, the uniform scalar quantizer is the optimal entropy coding quantizer in rate-distortion performance [19,20]. In other words, for the BCS measurements, the rate-distortion performance of the uniform or non-uniform scalar quantization methods will be inferior to the joint performance of uniform quantization and entropy coding. Therefore, in current research on CS for images, the CS measurements are usually quantified using the uniform scalar quantization method [21], and the quantized measurements are encoded by entropy coding to improve the compression performance [22,23]. However, since the computational cost involved in entropy coding is usually high [23,24], using entropy coding will reduce the low-complexity advantage of the CS encoder.

There are two main ways to improve rate-distortion performance. The first way is to reduce the bitrate while keeping the distortion constant, while the second way is to reduce the distortion while keeping the bitrate constant. Using entropy coding on quantized data is the first way. Without considering entropy coding, the second way is the only method to improve rate-distortion performance. Moreover, there are strategies aimed at enhancing the compression efficiency of CS. For example, ref. [25] introduces a novel application of Zadoff–Chu sequences, renowned for their excellent autocorrelation properties. By utilizing this sequence in the measurement matrix, sparsity in the compressive domain is enhanced, leading to improved recovery accuracy for quantized CS data. Ref. [26] explores the use of Discrete Fourier Transform (DFT) for measurement, enabling parallel processing capabilities and enhancing the efficiency of block compressive sensing. This strategy capitalizes on the computational advantages of FFT to accelerate the CS process while maintaining reconstruction fidelity. While many approaches can improve compression efficiency at the encoding stage [25,26,27,28], they are not necessarily applicable or effective for QCS.

While the field of block compressed sensing (BCS) has witnessed significant advancements in recent years, traditional quantization techniques employed in image/video coding applications continue to face pivotal challenges. Specifically, uniform quantization, a common choice due to its simplicity and compatibility with entropy coding, often incurs substantial quantization errors, compromising the fidelity of the reconstructed images. Moreover, it tends to overlook the inherent structure and correlations within the data, leading to encoding redundancy and inefficiency. On the other hand, Lloyd–Max quantization, although theoretically optimal for minimizing mean squared error, necessitates computationally intensive offline training and struggles to adapt dynamically to varying image characteristics. Furthermore, the reliance on entropy coding as a supplementary step to mitigate the loss from quantization adds to the computational burden and complexity of the encoding process.

In light of these challenges, our work introduces a novel convolutional neural network (CNN)-based quantization method specifically designed to overcome the drawbacks of traditional approaches. By leveraging the power of deep learning, our method transcends the uniformity imposed by classic quantizers, achieving a more nuanced mapping of measurements that closely follows their underlying distribution. This adaptive quantization, coupled with a sophisticated dequantization mechanism that harnesses correlation from quantized data, significantly reduces quantization errors without resorting to additional entropy coding.

In this paper, we propose a quantization method of BCS measurements that can reduce distortion while maintaining the bitrate. At the end of encoding, the proposed method models the measurements’ distribution of each image block. Subsequently, it quantizes the measurements of data that conform to a uniform distribution based on the distribution model. The proposed method uses the distribution parameters of each image block as side information of the encoder and adopts the same strategy to quantize the side information with 1 bit. At the end of decoding, the proposed method first restores the quantized data to data that conform to the distribution of the original measurements, then extracts the correlation information of the measurements from the quantized data of adjacent blocks to correct the measurements. Before the dequantization of the measurements, the same strategy is used to dequantize the side information. All quantization and dequantization processes are implemented as convolutional neural networks (CNN), and all networks are jointly learned offline. The CS coding structure based on the proposed method is shown in Figure 1.

The main contributions of this article are as follows:A quantization method of BCS measurement based on CNN is proposed. The proposed method constructs and jointly trains the CNN of the quantization and dequantization processes, which aims to maximize the coding output entropy and minimize the quantization error.A quantization process based on measurements’ distribution is proposed. Based on the properties of the cumulative distribution function (CDF), a neural network model was constructed to map measurements to the quantized data following a uniform distribution, which maximizes the amount of information carried in the quantized data. An activation function with a constrained output range was designed to reduce the computational complexity of the network’s activation function.A dequantization process based on the neighborhood information of measurements is proposed. The inverse process of quantization is used as a module to restore the quantized data to data that conform to the distribution of the original measurements. Furthermore, an information correction module is introduced to extract correlation information from multiple quantized values for correcting the measurements. The two modules are used to improve the quality of the dequantized measurements through residual connection.The distribution parameters of the block measurements are used as side information, which is quantized with 1 bit by the same quantization process.

While conventional approaches such as uniform quantization and Lloyd–Max quantization with entropy coding have been widely employed, they often introduce significant quantization errors and encoding inefficiencies. Our work diverges from these methodologies by introducing a CNN-based quantization strategy that not only maps measurements to a uniformly distributed quantized space to maximize the information content but also incorporates a novel dequantization process that leverages correlations from the quantized data to minimize reconstruction errors. This innovative method bypasses the need for entropy coding, offering a more efficient and adaptive solution for BCS applications.

In comparison to uniform quantization, which assigns equal intervals to the entire dynamic range, and Lloyd–Max quantization, known for minimizing the mean squared error but requiring complex optimization, our CNN-based approach dynamically adapts to the underlying data distribution. Unlike entropy coding methods, which reduce redundancy at the expense of computational complexity, our method directly optimizes the quantization process through learning, achieving superior performance without additional encoding steps.

The core of our method lies in the design of the CNN architecture, which jointly learns the quantization and dequantization processes. This contrasts with traditional quantization techniques that rely on predetermined, static decision boundaries. Our network specifically tailors the quantization levels based on the input data’s statistical properties, ensuring a closer match to the original signal characteristics. Additionally, the utilization of correlation information from the quantized data for post-processing further distinguishes our method, leading to reduced quantization artifacts.

The rest of this paper is organized as follows. Section 2 introduces the proposed method, which mainly includes the BCS quantization process, parameter estimation, parameter quantization and dequantization, and the BCS dequantization process. Section 3 presents the experimental results. The conclusion is given in Section 4.

## 2. Proposed Method

In this paper, we aim to improve the amount of information of quantized data and the information extraction ability of the dequantization process. The proposed method mainly includes the following aspects: (1) BCS quantization process, (2) parameter estimation, (3) parameter quantization and dequantization, and (4) BCS dequantization process.

### 2.1. BCS Quantization Process

Ref. [29] demonstrates that a random variable’s CDF can transform its data distribution into a uniform distribution. The random variables are usually modeled by probability density functions (PDF) to describe their distributions. It is difficult to obtain a closed-form expression for the CDF due to the need for integral calculations involving the PDF. Due to the strong function approximation ability of feedforward neural networks, we propose utilizing a feedforward neural network to model the CDF of the CS measurements.

Assuming that P(y) and F(y) are the PDF and CDF of the measurement variable y, respectively, P(y) and F(y) need to satisfy the following conditions:(1)$$\left\{\begin{array}{l} F(y)\geq 0\\ F(y)\leq 1\\ P(y)=\frac{\partial F(y)}{\partial x}\geq 0 \end{array}\right.$$

Compared with a three-layer feedforward neural network, a four-layer feedforward neural network can accurately establish the relationship between input and target variables with fewer hidden neurons [30]. When employing a four-layer feedforward neural network to build the CDF of the measurements, it can be represented as:(2)u1=yu2=g1W1u1+d1u3=g2W2u2+d2F(y)=g3W3u3+d3
where g1,g2,g3 represent the activation functions, and W1∈ℝL1×1, W2∈ℝL2×L1, W3∈ℝ1×L2, d1∈ℝL1×1, d2∈ℝL2×1, d3∈ℝ1×1 denote the model parameters.

Based on Equation (1), it is necessary to ensure that the output values of F(y) fall within [0, 1]. In model (2), the activation function $g_{3}$ determines the output range of F(y). Among the commonly used activation functions, the sigmoid function can ensure that the output values fall within [0, 1]. However, the sigmoid is a highly complex nonlinear activation function. We propose a rectified linear activation function with a limited output range, which can be expressed as:(3)$$G(x)=\left\{\begin{array}{ll} 1 & x\geq \alpha \\ \frac{x}{2\alpha }+\frac{1}{2} & -\alpha \leq x\leq \alpha \\ 0 & x\leq -\alpha \end{array}\right.$$
where α is a finite constant greater than 0. The curve graph and gradient curve of G(x) are shown in Figure 2.

To improve the adaptiveness of the activation function, we take α as a learnable parameter of the CDF model. Because activation functions usually do not come with trainable parameters, we transform Equation (3) as:(4)$$g_{3}(z)=\left\{\begin{array}{ll} 1 & z\geq 1\\ \frac{z}{2}+\frac{1}{2} & -1\leq z\leq 1\\ 0 & z\leq -1 \end{array}\right.$$

By comparing Equations (3) and (4), we obtain $G(x)=g_{3}\left(\frac{x}{\alpha }\right)$. Let $\beta =\frac{1}{a}$, and then the neural network model of CDF can be expressed as:(5)$$F(y)=g_{3}\left(\left(\beta \left(W_{3}g_{2}\left(\left(W_{2}g_{1}(W_{1}y+d_{1})+d_{2}\right)+d_{3}\right)\right)\right)\right)$$
where β∈ℝ, and $g_{1},g_{2}$ are the LeakyReLU activation function [31].

In BCS, all measurements are usually stored in the form of a matrix, which can be expressed as:(6)Y=y1,…,yNB=Φx1,…,xNB
where Y∈ℝM×NB represents the measurements’ matrix, Φ represents the measurement matrix, M represents the number of measurements for each image block, NB represents the number of image blocks, xi represents the pixel value vector of the i-th block, and yi represents the measurements’ vector of the i-th block.

When the input becomes a matrix, we can convert the feedforward neural network to a CNN. The number of convolutional layers in the CNN equals the number of parameter layers in the feedforward neural network. The number of channels in the convolutional network is equal to the number of neurons in the hidden layer. Since the feedforward neural network model has a single input and output, the kernel size of the convolutional network is 1 × 1. Considering both quality and complexity, we set the output feature of the intermediate convolutional layer to six channels. The network structure diagram of CNN based on model (5) is shown in Figure 3.

In Figure 3, “Conv” denotes convolutions, the numbers above “Conv” indicate the kernel size, “LeakyReLU” and “$g_{3}$” represent the activation functions used in the current convolutional layer, and the rectangular boxes represent the output feature maps of the convolutional layer. The numbers below the rectangular boxes indicate the number of channels. To balance the computational cost and fitting performance of the model (5), we adopt $L_{1}=L_{2}=6$ in this paper.

Due to variations between image blocks, the distribution parameters and CDF of different block measurements also differ. However, the model (5) does not account for the influence of distribution parameters, which limits its adaptability.

When using a Gaussian measurement matrix, the measurements of the same image block approximately follow a Gaussian distribution. Assuming that the measurement variable $y_{j}$ of the j-th image block follows a Gaussian distribution $N(y_{j};\mu_{j},\sigma_{j})$, it can be transformed into a standard normal distribution N(z;0,1) through Equation (7), which can be expressed as:(7)$$z=\frac{y_{j}-\mu_{j}}{\sigma_{j}}$$
where $\mu_{j}$ and $\sigma_{j}$ represent the location parameter (mean) and scale parameter (standard deviation), respectively. For the measurements’ matrix Y, Equation (7) can be expressed as:(8)$$Z=\left(Y-U\right)./\Lambda$$
where $U=\left[\left\{\mu_{j}\right\}\right]$ represents the matrix of position parameters, $\Lambda =\left[\left\{\sigma_{j}\right\}\right]$ represents the matrix of scale parameters, and ./ represents the element-wise division of matrices.

After removing the impact of distribution parameters through Equation (8), the same CNN can transform the measurements of all blocks into a uniform distribution.

In order to reduce the computational burden of parameter extraction, the matrix Y0∈ℝM0×NB of a small number of measurements is used for extracting the distribution parameters. The proposed BCS quantization process is shown in Figure 4. In Figure 4, “Repmat” represents repeat copies of an array, which is used to copy the parameters into the same size as the measurements’ matrix. Because the measurements of an image block have the same position and scale parameters, the size of the position and scale parameters are 1×NB.

### 2.2. Parameter Estimation

In general, the expressions of the mean (position parameter) μ and variance (scale parameter) $\sigma^{2}$ are as follows:(9)$$\begin{array}{l} \mu_{j}=\frac{\sum_{i}y_{j,i}}{N}\\ \sigma_{j}^{2}=\frac{1}{N}\sum_{i}{\left(y_{j,i}-\mu \right)}^{2} \end{array}$$

Since using partial measurements to estimate distribution parameters may introduce some errors, we use a neural network to estimate the distribution parameters. According to the definition of convolution, Equation (9) can be expressed as:(10)$$\begin{array}{l} \mu_{j}=\frac{\sum_{i}y_{j,i}}{N}=[y_{j,1},y_{j,2},\ldots ]*W_{u}\\ \sigma_{j}^{2}=\frac{1}{N}\sum_{i}{\left(y_{j,i}-\mu \right)}^{2}\\ =[\left([y_{j,1},y_{j,2},\ldots ]-[y_{j,1},y_{j,2},\ldots ]*W_{\mu }\right)\\ ⊙\left([y_{j,1},y_{j,2},\ldots ]-[y_{j,1},y_{j,2},\ldots ]*W_{\mu }\right)]*W_{\mu } \end{array}$$
where ∗ denotes the convolution operation, ./ represents the element-wise multiplication of matrices, and $W_{\mu }=\left[\frac{1}{N},\ldots ,\frac{1}{N}\right]$ denotes the convolution kernel used for computing the mean.

Equation (10) shows that the mean and variance can be estimated using convolutions. Based on Equation (10), we construct CNNs to estimate the position and scale parameters, as shown in Figure 5 and Figure 6, respectively. Since the mean and standard deviation functions are relatively simple in form, three channels are used for the middle convolution layer.

### 2.3. BCS Dequantization Process

The information of the quantized data is not only present in a single quantized value but also exists among multiple quantized values. Typically, the dequantization process at the decoding end is the inverse of the quantization process at the encoding end. However, the proposed quantization process and its inverse process only operate on individual quantized values. If dequantization only utilizes the inverse process of the proposed quantization process, it cannot use the correlated information among the multiple quantized values. Therefore, we propose adding a measurements’ information correction module to extract the measurement correction from the multiple neighboring quantized values. The dequantization process is shown in Figure 7. 

The quantization process is built based on the CDF and its inverse process can be constructed with the inverse function of the CDF. We adopt the same network architecture for the inverse process module. Since the similarity between adjacent image blocks is significant, we use convolution kernels of 1 × 3 to extract the compensation information from the neighborhood measurements. In order to improve the information extraction, five convolution layers are used in the measurement compensation information module and the inverse process of the proposed quantization. The network structure of the inverse process of the proposed quantization is shown in Figure 8. The network structure of the measurement compensation information module is illustrated in Figure 9.

### 2.4. Parameter Quantization and Dequantization

Due to the high similarity between adjacent image blocks, the distribution parameters exhibit some similarity. Therefore, we use the same quantization and dequantization processes to quantize and dequantize the distribution parameters. To reduce the extra bits of the side information, we use 1 bit to quantize the side information. The network structures of quantization and dequantization for the parameters are shown in Figure 10 and Figure 11, respectively.

### 2.5. Local Normalization and Loss Function

#### 2.5.1. Local Normalization of Measurements’ Matrix

The element in the i-th row and j-th column of the measurements’ matrix can be expressed as:(11)Yi,j=Φixj=∑k=1NΦi,kxk,j
where Φi represents the i-th row of the measurement matrix Φ. The grayscale values of the images are typically represented by 256 levels, the pixel values satisfy the following:(12)0≤xk,j≤255,xk,j∈ℤ

Combining Equations (11) and (12), we have:(13)$$255\sum_{k=1}^{N}\min\left\{\Phi_{i,k},0\right\}\leq \sum_{k=1}^{N}\Phi_{i,k}x_{k,j}\leq 255\sum_{k=1}^{N}\max\left\{\Phi_{i,k},0\right\}$$

Based on Equation (13), we take $255\sum_{k=1}^{N}\min\left\{\Phi_{i,k},0\right\}$ and $255\sum_{k=1}^{N}\max\left\{\Phi_{i,k},0\right\}$ as the minimum and maximum values of $Y_{i,j}$, which can be expressed as:(14)$$\begin{array}{l} Y_{i,j}^{max}=255\sum_{k=1}^{N}\max\left\{\Phi_{i,k},0\right\}\\ Y_{i,j}^{min}=255\sum_{k=1}^{N}\min\left\{\Phi_{i,k},0\right\} \end{array}$$

Based on Equation (14), each measurement can be normalized, which can be expressed as:(15)$${\tilde{Y}}_{i,j}=\frac{Y_{i,j}-Y_{i,j}^{min}}{Y_{i,j}^{max}-Y_{i,j}^{min}}$$

According to Equation (14), it is known that the same row of Y shares the same maximum and minimum values. Therefore, Equation (15) is referred to as the local normalization method for the measurements’ matrix.

Since Equation (14) only requires computation based on the measurement matrix, the row normalization method does not require transmitting the maximum and minimum value. Equation (15) transforms all measurements into real numbers between 0 and 1, and the input and output of the CNNs in the quantization and the dequantization processes are also real numbers between 0 and 1. Therefore, the quantization of the measurements’ matrix can be expressed as:(16)$$Q\left[Y\right]=\mathrm{Round}\left(F_{q-net}\left(\tilde{Y}\right)\left(2^{b}-1\right)\right)$$
where *b* represents the bit-depth of quantization, and $F_{q-net}$ represents the CNN in the quantization process.

At the decoding end, the output value of the dequantization network needs to be denormalized, which can be represented as:(17)$$\hat{Y}=F_{dq-net}\left(Q^{-1}\left[Q\left[\left(F_{q-net}\left(\tilde{Y}\right)\right)\right]\right]\right)\left(Y_{i,j}^{max}-Y_{i,j}^{min}\right)+Y_{i,j}^{min}$$
where $F_{dq-net}$ represents the CNN in the dequantization process.

#### 2.5.2. Loss Function

The main objective of the proposed method is to optimize the amount of information retained in the quantized data while minimizing the quantization error. The average information of data is usually measured by information entropy. However, since the information entropy depends on probability statistics, it cannot propagate the gradient. In this paper, we propose a continuous function to estimate the information entropy when training the network.

The entropy of the quantized measurements’ matrix $\overset{⌢}{Y}$ can be represented as follows:(18)$$H(\overset{⌢}{Y})=-\sum_{k}P(s_{k})\log_{2}\left(P(s_{k})\right)$$
where $P(s_{k})$ represents the probability of the symbol $s_{k}$ in $\overset{⌢}{Y}$, which requires counting the number of occurrences of symbol $s_{k}$.

We can count the numbers greater than $s_{k}$ in the measurements’ matrix by using a step function, which is defined as follows:(19)$$h\left(x\right)=\left\{\begin{array}{cc} 1 & x\geq 0\\ 0 & x<0 \end{array}\right.$$

The numbers greater than $s_{k}$ in the measurements’ matrix can be represented as:(20)$$\sum_{i}\sum_{j}h\left({\overset{⌢}{Y}}_{i,j}-s_{k}\right)$$

The step function $h\left(x\right)$ is not differentiable at 0. To ensure differentiability, we use the sigmoid activation function to approximate $h\left(x\right)$. The approximate function can be represented as follows:(21)$$h\left(x\right)\approx sigmoid(\eta x)$$

When η→+∞, we can get:(22)$$sigmoid\left(\eta \left({\overset{⌢}{Y}}_{i,j}-s_{k}+\varepsilon \right)\right)=\left\{\begin{array}{cc} 1 & {\overset{⌢}{Y}}_{i,j}\geq s_{k}\\ 0 & {\overset{⌢}{Y}}_{i,j}<s_{k} \end{array}\right.$$
where 0<ε<1 is used to ensure that the function maps to 1 when ${\overset{⌢}{Y}}_{i,j}=s_{k}$, and we set ε = 0.5. The parameter η can be used to adjust the variation of the sigmoid function. Based on practical experience, setting η=64 can make the output values of the sigmoid function approach 0 or 1 as much as possible. 

Based on Equation (22), the probability greater than $s_{k}$ in the measurements’ matrix can be represented as:(23)$$P(\overset{⌢}{Y}\geq s_{k})\approx \frac{\sum_{i}^{M}\sum_{j}^{N_{B}}sigmoid\left(\eta \left({\overset{⌢}{Y}}_{i,j}-s_{k}+0.5\right)\right)}{MN_{B}}$$

Assuming that $s_{k+1}>s_{k}$, we have:(24)$$\begin{array}{l} P(s_{k})\approx P(\overset{⌢}{Y}\geq s_{k})-P(\overset{⌢}{Y}\geq s_{k+1})\\ \approx \frac{\sum_{i}^{M}\sum_{j}^{N_{B}}sigmoid\left(\eta \left({\overset{⌢}{Y}}_{i,j}-s_{k}+0.5\right)\right)}{MN_{B}}-\frac{\sum_{i}^{M}\sum_{j}^{N_{B}}sigmoid\left(\eta \left({\overset{⌢}{Y}}_{i,j}-s_{k+1}+0.5\right)\right)}{MN_{B}} \end{array}$$

Based on the above, the computation of information entropy can be estimated as:(25)$$H(\overset{⌢}{Y})=-\sum_{k}\left(P(\overset{⌢}{Y}\geq s_{k})-P(\overset{⌢}{Y}\geq s_{k+1})\right)\log_{2}\left(P(\overset{⌢}{Y}\geq s_{k})-P(\overset{⌢}{Y}\geq s_{k+1})\right)$$

To simultaneously minimize the quantization error and maximize the information entropy, the objective function is composed of two parts. The first part is the mean square error (MSE) between the CS measurements before and after quantization, and the second part is the information entropy of the quantized measurements. The loss function is as follows: (26)$$Loss=MSE(Y,\hat{Y})-\lambda H(\overset{⌢}{Y})$$
where λ>0 is a parameter that controls the importance of the information entropy of quantized data. $\hat{Y}$ represents the dequantized measurements.

## 3. Results

In this section, we present various experimental results that validate the performance of our method. The proposed method is primarily implemented through CNNs, which requires the collection of a training dataset for network training. The training dataset comprises 200 training images taken from the BSDS500 dataset [32]. Each image has been cropped into grayscale images of size 256 × 256 with a stride of 60 pixels. The block size utilized in BCS is set at 16 × 16. The samples and labels of the training data set are both a matrix of BCS measurements for each image. The matrix used to collect the measurements has a sampling rate of 0.8, so the trained network can be applied to any measurements’ matrix with a sampling rate lower than 0.8. Each block typically requires at least ten measurements to reconstruct an image from the BCS measurements efficiently, so we take $M_{0}=10$ as the number of the partial measurements.

All CNNs were implemented using the Pytorch framework. We trained the CNNs of the quantization and dequantization processes together. The batch size was set to 32, with the optimization process performed using the Adam algorithm, initialized with a learning rate of 0.001. After the initial training of 10,000 epochs, the learning rate was reduced by a factor of 10, and all networks were trained for an additional 20,000 epochs. The training process was conducted on a server powered by an Intel Xeon CPU, a Nvidia RTX 2080Ti GPU with 11 GB of memory, and 128 GB of DDR4 RAM. The test images consisted of an APC, aerial, airplane, airport, building, moon surface, tank, and truck, as illustrated in Figure 12. Publicly available datasets such as Set5 [33] (5 images), Set11 [34] (11 images), Set14 [35] (14 images), and BSD68 [36] (68 images) were also employed. All experiments were conducted on an Intel Core i5-8300H CPU @ 2.30GHz, and the proposed method’s performance was measured using the PSNR of the reconstructed images.

### 3.1. Analysis of Measurement Reconstruction Results

In this section, we analyzed the number of reconstruction levels and the quantization errors of the quantization method. 

The current studies on improving CS quantization methods typically focus on sparse signals, but these methods are not suitable for images with a large number of elements. To ensure a low complexity of the encoder, the BCS encoder usually uses uniform quantization and entropy coding to process the BCS measurements. In addition, uniform quantizer is considered as the optimal quantizer for entropy-coded quantization in data compression theory [23,24], which is why BCS encoders tend to use uniform quantization and entropy coding. Currently, the most advanced quantization techniques for BCS of images are believed to be the prediction quantization method [9] and the progressive quantization method [10]. However, they essentially improve the coding strategy, while the quantization method employed is still uniform quantization, which can explore improvements using the commonly used quantization methods and the proposed method. To simplify the experimental process, this paper only compares the quantization techniques, using entropy-coded uniform quantization, µ-law quantization [37,38], and Lloyd–max quantization [39,40] as benchmarks. The entropy-coded uniform quantization method refers to the use of entropy coding after performing uniform quantization on the measurements. The codebook for Lloyd–Max quantization was obtained through offline training.

In scalar quantization methods, the number of reconstruction levels is typically equal to the number of quantization levels, as shown in Table 1. Table 2 shows the number of reconstructed levels of the proposed method for eight test images at a measurement rate of 0.2. Comparison between Table 1 and Table 2 shows that the proposed method have more different elements in the dequantized measurements. This is mainly because the proposed method utilizes the information of multiple quantized values for the dequantization, which gives the proposed method the advantage of many-to-many mapping. Moreover, each row of the measurements’ matrix adopts different maximum and minimum values for local denormalization. The local normalization approach also increases the number of reconstruction levels in accordance with the increase in measurement rate.

The greater the number of reconstruction levels of dequantized measurements, the greater the quantization error can be reduced. Table 3, Table 4 and Table 5 display the MSE of the various quantization methods for the measurements quantized with 3-bit, 6-bit, and 8-bit, respectively.

Table 3 shows that when using 3-bit quantization, the proposed method reduces the MSE by 788.07, 670.48, and 585.35 compared with uniform quantization, µ-law quantization, and Lloyd–Max quantization, respectively. Similarly, in Table 4, the proposed method accomplishes a reduced MSE by 8.25, 6.77, and 10.66 when 6-bit quantization is employed. Table 5 reveals that when 8-bit quantization is applied, the proposed method reduces the MSE by 0.4665, 0.3765, and 1.8162 compared with uniform quantization, µ-law quantization, and Lloyd–Max quantization, respectively. Table 3, Table 4 and Table 5 demonstrate that the proposed method has a significantly lesser MSE than other quantization methods.

### 3.2. Analysis of the Impact of Entropy Loss Constraints

In this section, we analyzed the effect of entropy constraint. The parameter λ in the loss function determines the degree of entropy constraint. In order to analyze the appropriate value of λ, we only select a few common values for training the CNNs of the quantization and dequantization processes. Table 6 shows the MSE of the dequantized measurements and the information entropy of the quantized measurements when 8-bit is used to quantize the measurements of the BSDS500 dataset.

It can be seen from Table 6 that the entropy constraint can increase the information entropy of the quantized measurements but it has a slight impact on reducing the MSE of the dequantized measurements. When λ=0.05, the MSE of the dequantized measurements is the smallest. Therefore, when training the proposed network, we use λ=0.05.

In addition, it can be observed that the entropies of the quantified measurements are very close to the bit-depth. Some images may not be compressed when using a fixed code table for entropy coding. In other words, the measurements quantized by the proposed method do not need entropy coding.

### 3.3. Analysis of the Impact of the Measurement Information Correction Module

In this section, we analyzed the impact of the information correction module on the dequantization process. Table 7 shows the quantization performance of the BSDS500 dataset when different information correction modules are used in the proposed method. In Table 7, Contrast Scheme 1 did not use a measurement information correction module. Contrast Scheme 2 used a measurement information correction module composed of three convolutional layers. Contrast Scheme 3 used a measurement information correction module composed of six convolutional layers.

Table 7 reveals that the proposed method has significant advantages over the µ-law quantization method. Compared with Contrast Scheme 1, the MSE of Contrast Scheme 2 is reduced by 0.0271, while its entropy is increased by 0.067. Similarly, the MSE of Contrast Scheme 3 is reduced by 0.0457 and the entropy is increased by 0.1616. These results illustrate that the information correction module effectively improves the quantization performance. Furthermore, the information correction module exhibits a stronger correction capability with the increase of convolutional layers.

### 3.4. Rate-Distortion Performance Comparison

In this section, we compared the rate-distortion performance of the proposed method with the entropy-coded uniform quantization, µ-law quantization, and Lloyd–Max quantization. The entropy-coded uniform quantization method refers to the use of entropy coding after performing uniform quantization on the measurements, which is expressed by “uniform quantization + entropy coding” in this paper. When drawing the rate-distortion curve, we traverse multiple quantization bit-depths and sampling rates to encode and decode the test images. Then, we choose the optimum Bitrate-PSNR points and connect them with a line. The bit-depth adopts seven values in {2, 3, 4, …, 8}, and the sampling rate chooses 77 values in {0.04, 0.05, 0.06, …, 0.8}. The image reconstruction algorithm used is the BCS-SPL-DCT algorithm [41]. When calculating the bitrate of “uniform quantization + entropy coding,” the average codeword length of entropy coding is replaced by information entropy. Figure 13 shows the PSNR curve of the eight test images.

In Figure 13, the proposed method has the best rate-distortion performance on all eight test images, particularly for the aerial, building, and tank images. The PSNR curve of “uniform quantization + entropy coding” is better than the µ-law and Lloyd–Max quantization methods. This observation confirms that the existing quantizers without entropy coding have inferior rate-distortion performance compared with “uniform quantization + entropy coding.”

Figure 14 shows the reconstructed images of the eight test images with different methods at a compression bit rate of 0.1. The Lloyd–Max quantization approach generates an adaptive quantization dictionary for each image. We do not count the bits of the quantization dictionary in the compression bit rate of the Lloyd–Max quantization method. Therefore, the results of the Lloyd–Max quantization method in Figure 14 are equivalent to the optimal results of the conventional quantizer.

As shown in Figure 14, the proposed method exhibits the best visual effect and PSNR, followed by “uniform quantization + entropy coding” and Lloyd–Max quantization. Compared with “uniform quantization + entropy coding,” for the eight test images, the PSNR of the proposed method increased by 0.65 dB, 0.44 dB, 1.97 dB, 0.02 dB, 0.46 dB, 0.09 dB, 0.37 dB, and 0.29 dB, respectively. Compared with the Lloyd–Max quantization, for the eight test images, the PSNR of the proposed method increased by 2.1 dB, 0.75 dB, 1.8 dB, 0.28 dB, 0.78 dB, 1.55 dB, 1.76 dB, and 1.53 dB, respectively.

The four quantization methods were also tested on the four test image datasets, and the average PSNR curves are shown in Figure 15.

In Figure 15, the average PSNR is the mean of the PSNR of the reconstructed images at a given bit rate for all images in the dataset. The PSNRs at a given bit rate are obtained by linear interpolation from the Bitrate-PSNR curve for each image. The given bit rates are set to {0.1, 0.2, …, 1 bpp}. For datasets Set5, Set11, Set14, and BSD68, the average PSNR curve of the proposed method is better than “uniform quantization + entropy coding,” µ-law quantization and Lloyd–Max quantization. Particularly, at a low bit rate (around 0.1 bpp), the proposed method’s PSNR is much higher than the other methods. 

The datasets SunHays80 [42] and Urban100 [43] have been extended for testing (all data are converted to grayscale images with 256 × 256). The quality of reconstruction is evaluated by the peak signal to noise ratio (PSNR) and the structural similarity (SSIM) between the reconstructed image and the original image. Table 8 shows the PSNRs and SSIMs of the four datasets at a bit rate of 0.1 bpp. Table 9 shows the PSNRs and SSIMs of the four datasets at a bit rate of 0.2 bpp. 

For all images of the six datasets, when the bitrate is set to 0.1 bpp, the proposed method, “uniform quantization + entropy coding,” µ-law quantization, and Lloyd–Max quantization achieve average PSNRs of 19.69 dB, 19.24 dB, 17.54 dB, and 18.67 dB, respectively. Compared with “uniform quantization + entropy coding,” the proposed method improves the PSNR by an average of 0.45 dB without entropy coding. The proposed method, “uniform quantization + entropy coding,” µ-law quantization, and Lloyd–Max quantization achieve average SSIMs of 0.1855, 0.1739, 0.1408, and 0.1547 respectively. Compared with “uniform quantization + entropy coding,” the proposed method improves the SSIM by an average of 0.0116 without entropy coding.

For all images of the six datasets, when the bitrate is set to 0.2 bpp, the proposed method, “uniform quantization + entropy coding,” µ-law quantization, and Lloyd–Max quantization achieve average PSNRs of 21.27 dB, 21.09 dB, 20.88 dB, and 20.93 dB, respectively. Compared with “uniform quantization + entropy coding,” the proposed method improves the PSNR by an average of 0.18 dB without entropy coding. The proposed method, “uniform quantization + entropy coding,” µ-law quantization, and Lloyd–Max quantization achieve average SSIMs of 0.2738, 0.2683, 0.2543, and 0.2567, respectively. Compared with “uniform quantization + entropy coding,” the proposed method improves the SSIM by an average of 0.0055 without entropy coding.

Across all images in the six datasets, the proposed method demonstrates superior performance compared to standard uniform quantization combined with “uniform quantization + entropy coding,” as well as the μ-law and Lloyd–Max quantization schemes.

### 3.5. Analysis of Computational Complexity

On the encoding side, the calculation of the proposed quantization method involves four networks: the position parameter estimation network, the scale parameter estimation network, the parameter quantization network, and the measurement quantization network. The network structure of the position parameter estimation network and scale parameter estimation network are identical, as shown in Table 10. Similarly, the network structures of the parameter quantization and measurement quantization are identical, as shown in Table 11.

The position and scale parameters are derived from partial measurements Y0∈ℝ10×NB. According to Table 10, convolutional layer 1 requires around 10×NB×3 multiplications and 10×NB×3 additions, convolutional layer 2 requires 3×NB×3 multiplications and 3×NB×3 additions, and convolutional layer 3 requires 3×NB multiplications and 3×NB additions. In total, location parameter estimation and scale parameter estimation need about 84NB times multiplications, 84NB times additions, and 12NB times LeakyReLU operations.

According to Table 11, to quantize a parameter or measurement, the quantization network typically requires 48 times multiplications, 48 times additions, 12 times LeakyReLU operations, and one time g3 operation. For the measurements’ matrix Y∈ℝM×NB, the numbers of measurements and parameters that need to be quantized are M×NB and 2NB, respectively. In total, it is necessary to compute (2+M)48NB times multiplications, (2+M)48NB times additions, (2+M)NB times LeakyReLU, and (2+M)NB times g3.

The proposed method requires approximately the same number of multiplication and addition operations, and the activation function to be computed only involves linear operations. Since addition is much faster than multiplication in practical operations, we only compared the number of multiplication operations. With an image size of 256 × 256 and a block size of 16 × 16, the total number $N_{B}$ of blocks would be 256. Assuming that the measurement rate of BCS is 0.1, each block obtains 26 measurements. Each measurement needs about 256 times multiplications and 255 times additions, and the calculation of measurements is about $26\times 256N_{B}$ times multiplications and $26\times 255N_{B}$ times additions. The proposed quantization method requires $660N_{B}$ times multiplications, $660N_{B}$ times additions, $156N_{B}$ times LeakyReLUs, and $12N_{B}$ times $g_{3}$. Compared with the calculation for BCS measurements, the calculation of the proposed quantization process is about 9.92% of that of the BCS measurements.

## 4. Conclusions

In this paper, we propose a BCS measurement quantization method based on CNNs. The method uses a CNN based on measurements’ CDF to map BCS measurements into quantized data following a uniform distribution. The block measurements’ distribution parameters are quantized as the side information of the encoder. In dequantization, a CNN based on information correction is designed by using the correlations between the BCS measurements. The proposed method uses partial measurements to extract parameter features and jointly optimizes the CNNs of the quantization and dequantization. The experimental results indicate that the proposed quantization method has better performance than joint uniform quantization and entropy coding.

In the future, investigating the performance of our method under adverse conditions, such as noisy or blurred inputs, is critical for real-world deployment. While the CNNs have shown remarkable performance, their computational and memory requirements are points for consideration. Exploring lightweight CNN architectures, pruning techniques, and hardware acceleration strategies would be instrumental in making our method more practical for resource-constrained devices.

## Figures and Tables

**Figure 1 entropy-26-00468-f001:**
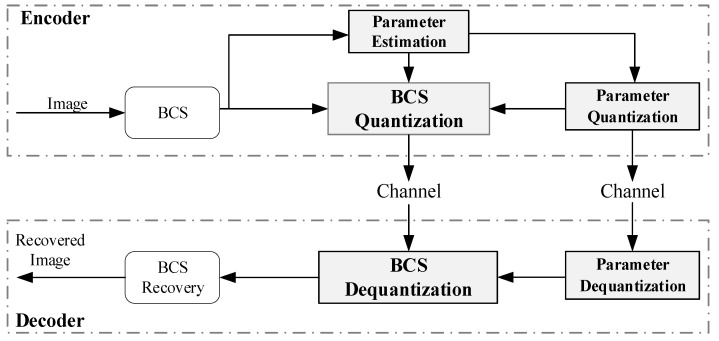
The CS coding structure based on the proposed method.

**Figure 2 entropy-26-00468-f002:**
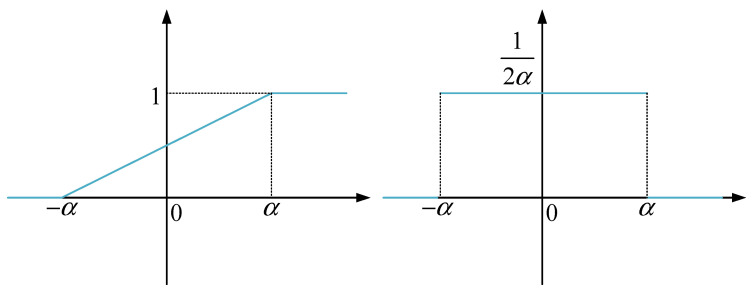
The curve and gradient curve of the proposed activation function.

**Figure 3 entropy-26-00468-f003:**
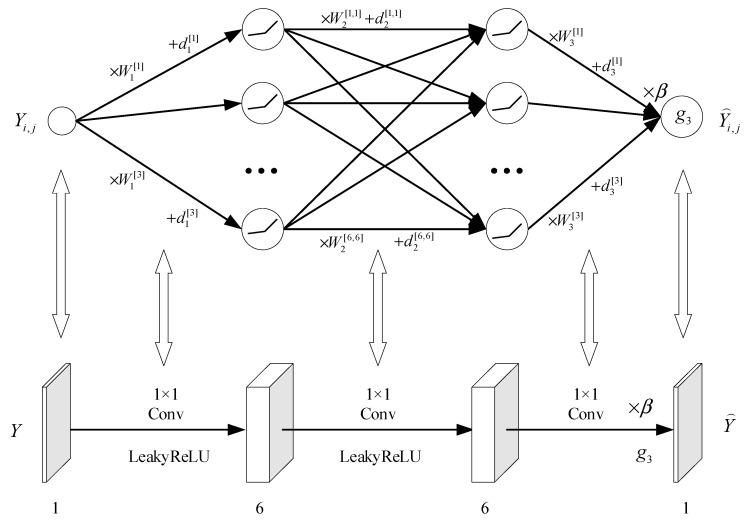
Network structure diagram of CNN based on model (5).

**Figure 4 entropy-26-00468-f004:**
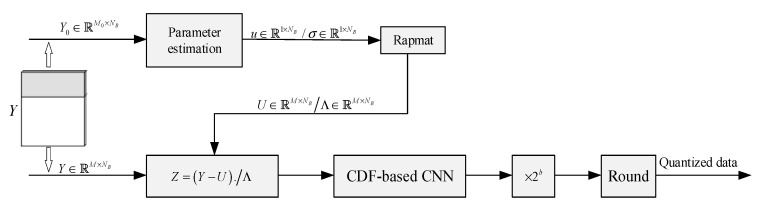
Structure diagram of the proposed BCS measurement quantization process.

**Figure 5 entropy-26-00468-f005:**
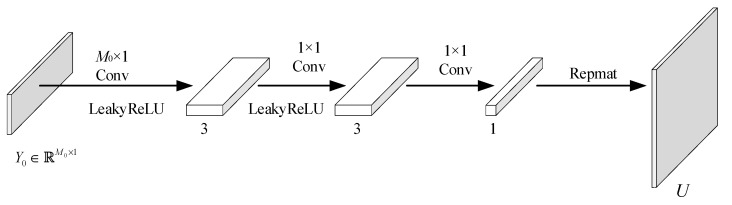
Network structure diagram for estimating the location parameters.

**Figure 6 entropy-26-00468-f006:**
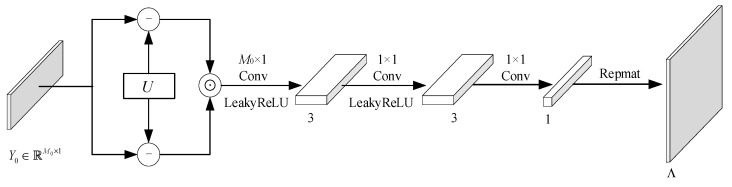
Network structure diagram for estimating the scale parameters.

**Figure 7 entropy-26-00468-f007:**
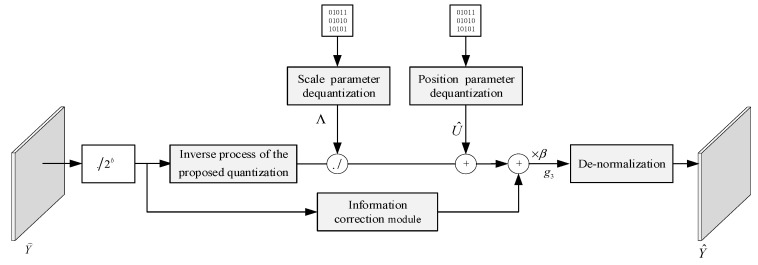
Structure diagram of the proposed dequantization process.

**Figure 8 entropy-26-00468-f008:**
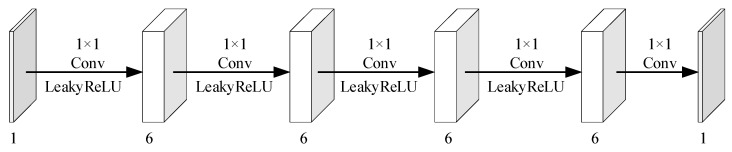
Network structure diagram of the inverse process of the proposed quantization.

**Figure 9 entropy-26-00468-f009:**
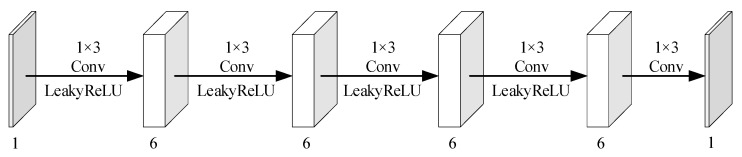
Network structure diagram of the measurement information correction module.

**Figure 10 entropy-26-00468-f010:**
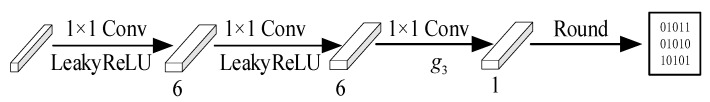
Network structure diagram for quantizing parameters.

**Figure 11 entropy-26-00468-f011:**
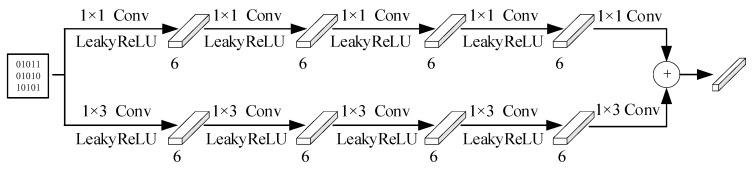
Network structure diagram for dequantizing parameters.

**Figure 12 entropy-26-00468-f012:**
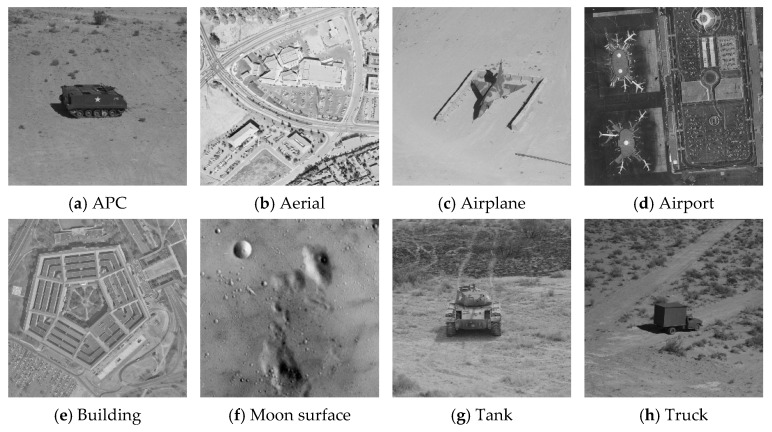
Eight test images.

**Figure 13 entropy-26-00468-f013:**
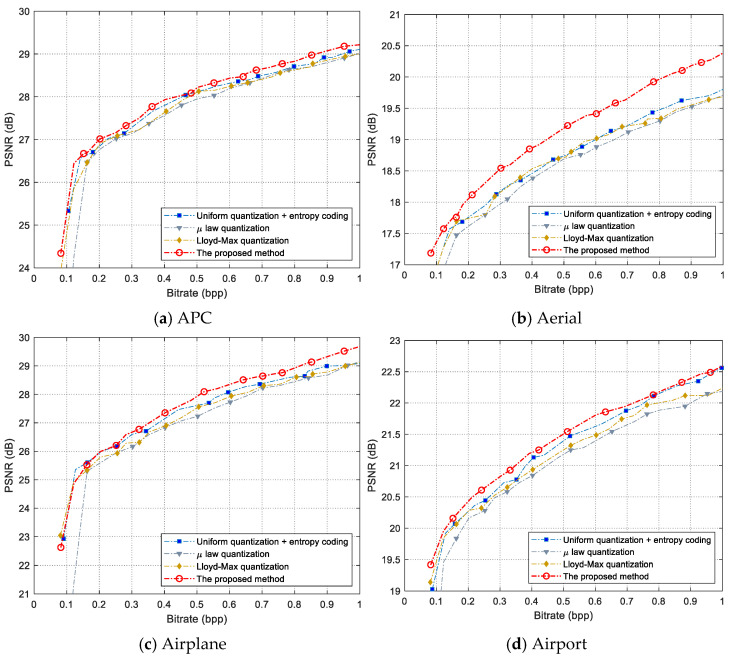
PSNR curves of the eight test images.

**Figure 14 entropy-26-00468-f014:**
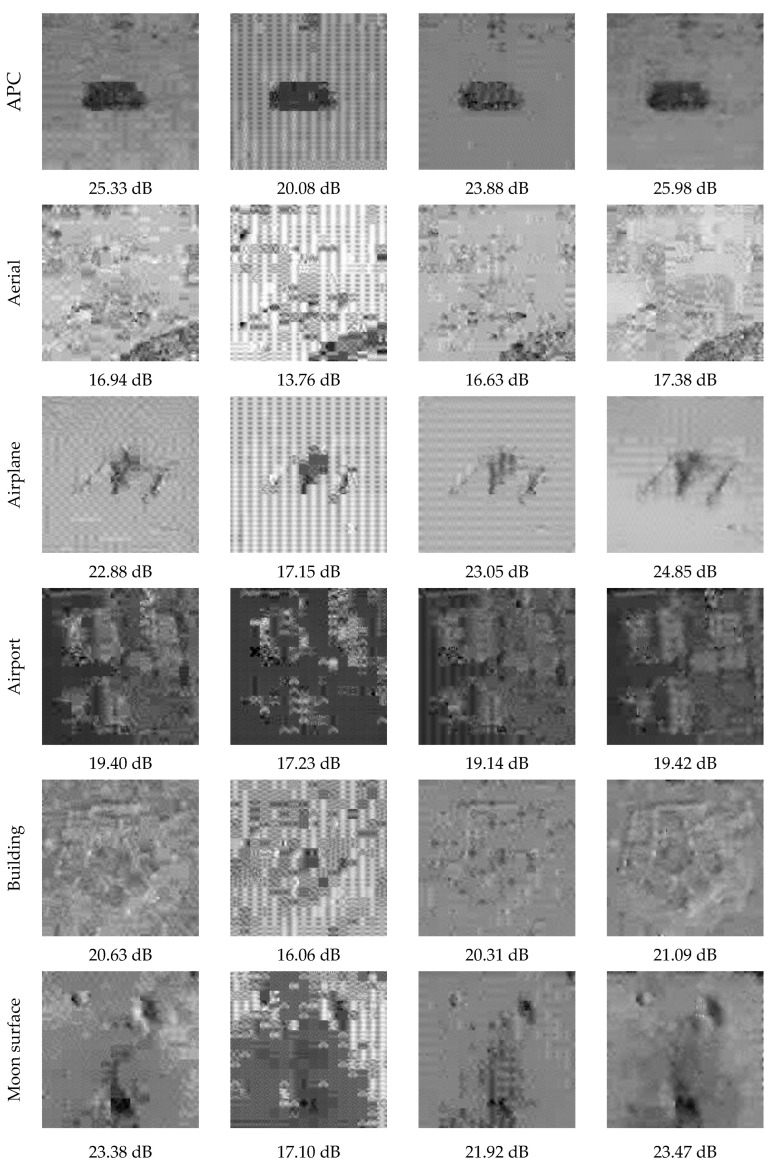
Visual comparison of different methods at a compression bit rate of 0.1.

**Figure 15 entropy-26-00468-f015:**
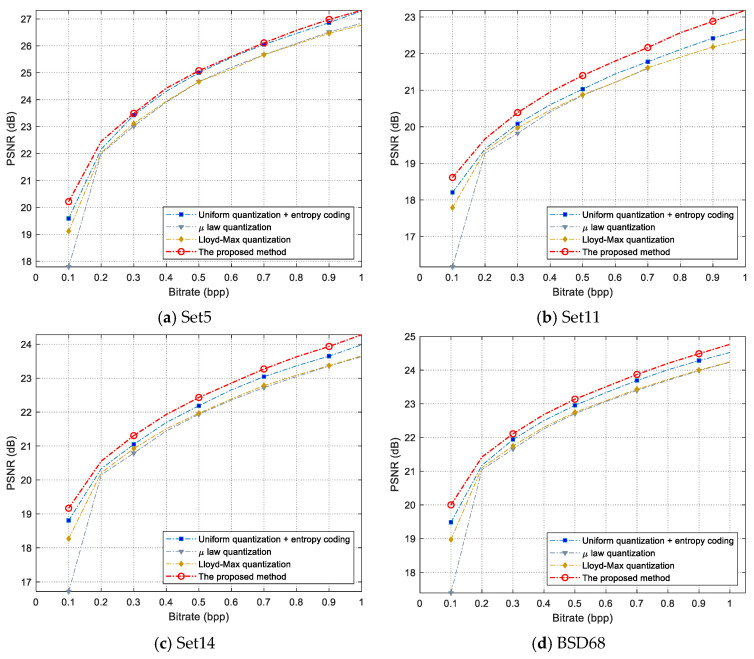
Average PSNR curves of test image sets.

**Table 1 entropy-26-00468-t001:** Number of reconstruction levels for scalar quantization methods.

Quantization Bit-Depth	2	3	4	5	6	7	8
Number of reconstruction levels	4	8	16	32	64	128	256

**Table 2 entropy-26-00468-t002:** Number of reconstruction levels for the proposed quantization.

Quantization Bit-Depth	2	3	4	5	6	7	8
APC	147	236	397	690	1147	1860	2921
Aerial	199	321	559	981	1712	2863	4368
Airplane	181	273	453	724	1122	1644	2336
Airport	186	283	488	845	1483	2545	4009
Building	161	243	414	739	1302	2277	3699
Moon surface	171	270	455	782	1328	2205	3515
Tank	161	241	413	744	1345	2422	3963
Truck	144	227	396	699	1220	2140	3527
Average	168.7	261.7	446.8	775.5	1332.3	2244.5	3542.2

**Table 3 entropy-26-00468-t003:** MSE of the measurements quantized with 3-bit.

Image	Uniform Quantization	µ-Law Quantization	Lloyd–Max	Proposed Method
APC	567.30	675.16	493.99	115.76
Aerial	1468.86	1250.97	1223.14	192.34
Airplane	1147.16	1280.45	1279.69	161.52
Airport	704.45	435.69	460.30	155.19
Building	864.87	768.95	629.43	113.59
Moon surface	1327.97	831.08	573.44	115.48
Tank	780.74	730.11	634.44	121.87
Truck	540.80	489.04	486.02	121.88
Average	925.27	807.68	722.56	137.20

**Table 4 entropy-26-00468-t004:** MSE of the measurements quantized with 6-bit.

Image	Uniform Quantization	µ-Law Quantization	Lloyd–Max	Proposed Method
APC	6.86	7.15	11.43	2.53
Aerial	17.77	16.11	21.37	3.87
Airplane	14.04	13.86	19.62	3.20
Airport	8.27	5.41	9.03	3.33
Building	11.02	9.78	12.89	2.51
Moon surface	15.11	10.68	11.66	2.58
Tank	9.61	8.58	12.32	2.63
Truck	6.60	5.88	10.21	2.63
Average	11.16	9.68	13.57	2.91

**Table 5 entropy-26-00468-t005:** MSE of the measurements quantized with 8-bit.

Image	Uniform Quantization	µ-Law Quantization	Lloyd–Max	Proposed Method
APC	0.4194	0.4337	1.8716	0.1742
Aerial	1.0692	1.0155	2.6658	0.2978
Airplane	0.8546	0.8304	2.5470	0.2356
Airport	0.5031	0.3274	1.5868	0.2543
Building	0.6684	0.5927	2.0241	0.1770
Moon surface	0.9308	0.6440	1.8995	0.1778
Tank	0.5884	0.5104	1.9121	0.1980
Truck	0.4032	0.3630	1.7278	0.1902
Average	0.6796	0.5896	2.0293	0.2131

**Table 6 entropy-26-00468-t006:** MSE and entropy for different λ.

λ	MSE	Entropy
0	0.2883	7.2624
0.05	0.2841	7.3376
0.1	0.2867	7.3539
0.2	0.2896	7.3787
0.5	0.3027	7.4487

**Table 7 entropy-26-00468-t007:** Quantization performance of different information correction modules.

	MSE	Entropy
µ-law quantification	1.0941	7.1510
Contrast Scheme 1	0.3298	7.1760
Contrast Scheme 2	0.3027	7.2430
Contrast Scheme 3	0.2841	7.3376

**Table 8 entropy-26-00468-t008:** The PSNRs of the four datasets at a bit rate of 0.1 bpp.

		Set5	Set11	Set14	BSD68	SunHays80	Urban100	Average
Uniform Quantization + Entropy Coding	PSNR	19.59	18.21	18.81	19.49	19.32	20.01	19.24
	SSIM	0.2117	0.1526	0.1769	0.1547	0.1361	0.2117	0.1739
µ-law Quantization	PSNR	17.79	16.17	16.72	17.40	17.30	19.86	17.54
	SSIM	0.1546	0.1176	0.1336	0.1080	0.1001	0.2311	0.1408
Lloyd–Max Quantization	PSNR	19.12	17.79	18.27	18.97	18.75	19.12	18.67
	SSIM	0.1909	0.1355	0.1587	0.1328	0.1196	0.1909	0.1547
Proposed Method	PSNR	20.22	18.61	19.17	20.00	19.90	20.25	19.69
	SSIM	0.2229	0.1672	0.1875	0.1636	0.1489	0.2229	0.1855

**Table 9 entropy-26-00468-t009:** The PSNRs of the four datasets at a bit rate of 0.2 bpp.

		Set5	Set11	Set14	BSD68	SunHays80	Urban100	Average
Uniform Quantization + Entropy Coding	PSNR	22.26	19.41	20.32	21.27	20.99	22.26	21.09
	SSIM	0.3246	0.2269	0.2653	0.2421	0.2260	0.3246	0.2683
µ-law Quantization	PSNR	22.00	19.28	20.15	21.05	20.84	21.94	20.88
	SSIM	0.3127	0.2100	0.2516	0.2275	0.2096	0.3141	0.2543
Lloyd–Max Quantization	PSNR	22.03	19.34	20.22	21.11	20.87	22.03	20.93
	SSIM	0.3127	0.2168	0.2542	0.2298	0.2139	0.3127	0.2567
Proposed Method	PSNR	22.43	19.6707	20.55	21.42	21.13	22.43	21.27
	SSIM	0.3249	0.2421	0.2725	0.2473	0.2313	0.3249	0.2738

**Table 10 entropy-26-00468-t010:** Detailed network structures of parameter estimation.

Convolution Layer	Kernel Size	Stride	Input Channels	Output Channels	Activation Function
Conv1	10 × 1	1	1	3	LeakyReLU
Conv2	1 × 1	1	3	3	LeakyReLU
Conv3	1 × l	1	3	1	-

**Table 11 entropy-26-00468-t011:** Detailed network structures of the quantization process.

Convolution Layer	Kernel Size	Stride	Input Channels	Output Channels	Activation Function
Conv1	1 × 1	1	l	6	LeakyReLU
Conv2	1 × 1	1	6	6	LeakyReLU
Conv3	1 × 1	1	6	1	$g_{3}$

## Data Availability

Data are contained within the article.

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
