# Peer review of "A Convolutional Neural Network-Based Quantization Method for Block Compressed Sensing of Images"

_entropy, 2024, doi:10.3390/e26060468_

Round 1
Reviewer 1 Report
Comments and Suggestions for Authors
The paper "A Convolutional Neural Network-Based Quantization Method for Block Compressed Sensing of Images" presents a novel quantization method for Block Compressed Sensing (BCS) measurements using Convolutional Neural Networks (CNNs). The authors address a significant challenge in resource-constrained image/video coding applications: the quantization of BCS measurements, which leads to substantial errors and encoding redundancy. Before publication, the authors should address the following comments in the manuscript:
1) The paper presents a novel method; however, the distinction between this work and the closest existing solutions in the literature could be more clearly emphasized.
2) It would be beneficial to discuss the limitations of existing methods that the proposed solution overcomes.
3) The literature review part of the paper needs expansion. The authors are advised to incorporate relevant state-of-the-art research studies:
DPCM for quantized block-based compressed sensing of images
Evolutionary multiobjective multiple description wavelet based image coding in the presence of mixed noise in images
An Image-Based Quantized Compressive Sensing Scheme Using Zadoff–Chu Measurement Matrix
Dynamic Multiple Description Wavelet Based Image Coding Using Enhanced Particle Swarm Optimization
Image parallel block compressive sensing scheme using DFT measurement matrix
4) While the experimental results demonstrate the effectiveness of the proposed method, the selection of datasets, benchmarks, and comparison metrics could be expanded. Including more diverse datasets and comparison with state-of-the-art methods using additional metrics could provide a more comprehensive evaluation of the proposed method's performance.
5) While the paper concludes with a brief mention of potential future work, a more detailed discussion on the future direction would be valuable.
Reviewer 2 Report
Comments and Suggestions for Authors
The authors proposed a new method for recoverying data from quantized Block Compressed Sensing (BCS) measurements using Convolutional Neural Networks (CNN).
The results look novel and interesting, so I recommend its publication in Entropy after a minor revision.
Some comments:
- A commonly used reference for Compressed Sensing is:
S. Foucart and H. Rauhut, A Mathematical Introduction to Compressive Sensing, Basel, Birkhäuser, 2013.
I suggest the authors to include this book into the reference list along with those original papers by Candes and Donoho published in 2006 [1-3].
- p. 4, line 135:
"\beta = \frac{x}{\alpha}" → "\beta = \frac{1}{\alpha}"
- p. 4, line 137:
"where $\beta \in \mathbb{R}^{1 \times 1}$. $g_1$ and $g_2$ use the LeakyReLU activation function"
→ "where $\beta \in \mathbb{R}$, and $g_1$ and $g_2$ are the LeakyReLU activation function"
- p. 4, line 140:
"the measurements' matrix" → "the matrix of measurements"
Round 2
Reviewer 1 Report
Comments and Suggestions for Authors
The authors propose a novel and important quantization method for BCS measurements utilizing convolutional neural networks (CNNs), which makes an important contribution to the literature. Moreover, they have extensively revised the manuscript based on my feedback. Therefore, I recommend the acceptance of the paper.